# Sodium Dodecyl Sulfate Analogs as a Potential Molecular Biology Reagent

Tsutomu Arakawa [1,*], Takako Niikura [2], Yoshiko Kita [1] and Teruo Akuta [3]

1 Alliance Protein Labs, 13380 Pantera Rd., San Diego, CA 92130, USA; kita36@gmail.com
2 Department of Information and Communication Sciences, Faculty of Science and Technology, Sophia University, Tokyo 102-8554, Japan; niikura@sophia.ac.jp
3 Research and Development Division, Kyokuto Pharmaceutical Industrial Co., Ltd., 3333-26, Aza-Asayama, Kamitezuna, Takahagi-shi 318-0004, Japan; t.akuta@kyokutoseiyaku.co.jp
* Correspondence: tarakawa2@aol.com

**Abstract:** In this study, we review the properties of three anionic detergents, sodium dodecyl sulfate (SDS), Sarkosyl, and sodium lauroylglutamate (SLG), as they play a critical role in molecular biology research. SDS is widely used in electrophoresis and cell lysis for proteomics. Sarkosyl and, more frequently, SDS are used for the characterization of neuropathological protein fibrils and the solubilization of proteins. Many amyloid fibrils are resistant to SDS or Sarkosyl to different degrees and, thus, can be readily isolated from detergent-sensitive proteins. SLG is milder than the above two detergents and has been used in the solubilization and refolding of proteins isolated from inclusion bodies. Here, we show that both Sarkosyl and SLG have been used for protein refolding, that the effects of SLG on the native protein structure are weaker for SLG, and that SLG readily dissociates from the native proteins. We propose that SLG may be effective in cell lysis for functional proteomics due to no or weaker binding of SLG to the native proteins.

**Keywords:** SDS; Sarkosyl; sodium lauroyl glutamate; cell lysis; refolding; neuropathological



## 1. Introduction

Detergents are one of the most widely used reagents in molecular biology research [1–3]. Their use includes but is not limited to cell lysis, electrophoresis, protein refolding, and the suppression of macromolecular surface adsorption or aggregation, i.e., a manipulation of macromolecular solutions [4]. Sodium dodecyl sulfate (SDS, see Figure 1 for chemical structure) is perhaps the most highly used reagent because of its use in SDS-polyacrylamide gel electrophoresis (SDS-PAGE). There are many advantages to the use of SDS in SDS-PAGE. SDS disrupts almost all the non-covalent molecular interactions and thereby allows the separation of individual components and the determination of their molecular weights [5]. SDS is effective at disrupting molecular interactions, including the lipid–lipid interactions that make cell membranes, and hence in releasing cytoplasmic and nuclear components [6]. This ability to disrupt molecular interactions is due to the strong micellar binding of SDS to the proteins, which provides the following advantage of using SDS in SDS-PAGE [7]. The SDS–protein complex in the gel can be readily electrophoresed to membranes in Western blotting technology [8,9]. However, such effects of SDS on disrupting molecular interactions means it can inherently denature macromolecules and disrupt their complexes, thereby destroying their functions.

There are two SDS analogs that are frequently used in protein or molecular biology research. As shown in Figure 1, they are sodium lauroly glutamate (SLG) and sodium lauroyl sarcosince (Sarkosyl). These are identical to SDS in their aliphatic portion and differ from SDS in the charged group. SLG has been used for cosmetic and cleaning applications [10]. Sarkosyl and SDS are consistently used for separating soluble and insoluble neuropathological fibrillar proteins, including tau, which is one of the microtubule-associating proteins

(MAPs) [11–13]. We examine the physical properties of these detergents and their mechanism in affecting protein structure, folding, and molecular interactions.

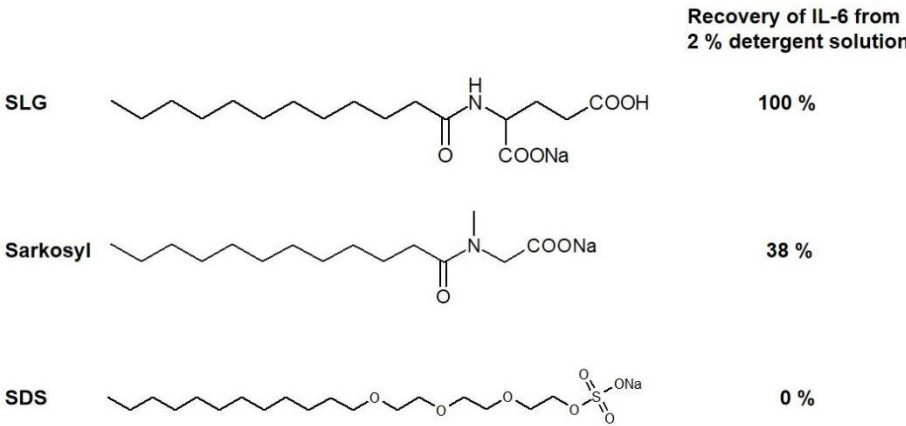

**Figure 1.** The structures of sodium lauroyl glutamate (SLG), Sarkosyl, and SDS. The recovery of interleukin-6 (L-6) from 2% detergent solutions is given in the last column. Native IL-6 was exposed to 0–2% detergent solutions and subjected to Superdex 75 gel filtration. The recovery of IL-6 was estimated from the absorbance of the protein eluted in the native IL-6 elution position.

## 2. Properties of SDS, Sarkosyl and SLG

Table 1 summarizes the properties of sodium dodecyl sulfate (SDS), sodium lauroyl sarcosine (Sarkosyl), and sodium lauroyl glutamate (SLG). Sarkosyl is unique in that it has a small micellar size compared to SDS and SLG, clearly forming a micellar structure with an aggregation number of ~80, as schematically depicted in Figure 2, in which red circles, yellow ellipsoids, and green squares depicted the different head groups of SDS, Sarkosyl, and SLG [13,14]. Sarkosyl is reported to have an aggregation number of two and, hence, does not appear to form a true micelle, although how this aggregation number is determined is not described [4]. There is no explanation about the stability of dimeric Sarkosyl: it may be worth mentioning that sodium caprylate with eight carbon aliphatic chains has a high monomeric concentration of 300 mM before forming micelles, meaning that the non-polar aliphatic chain can be stable in water. However, whether Sarkosyl forms a true micelle or not is questionable, and the critical micelle concentration (CMC) was reported to be 9.5–15 mM. Whether this corresponds to a CMC or a dissociation constant of the dimeric Sarkosyl (aggregation number 2) to a monomer may depend on how the CMC is determined. If it is determined from the surface tension measurement, then it simply refers to the equilibrium concentration of Sarkosyl between the air–water interface and the bulk solution. In fact, one of the CMC measurements is based on the air–water or solid–water interface adsorption of Sarkosyl [15] but not the actual equilibrium between micelles and free Sarkosyl in the bulk. The same authors claim that Sarkosyl is non-denaturing to proteins without showing data or references and may bind to proteins as assembled states, such as in the air–water interface.

**Table 1.** Properties of SDS, Sarkosyl, and sodium lauroyl glutamate (SLG).

| Detergent | CMC | Aggregation Number | MW |
|---|---|---|---|
| SDS | 8.2 mM in water 1.4 mM in 0.1 M NaCl | ~80 in water | 288.4 |
| Sarkosyl | 14.57 mM | 2 | 293.4 |
| SLG | 10.6 mM | ~80 | 351.4 |

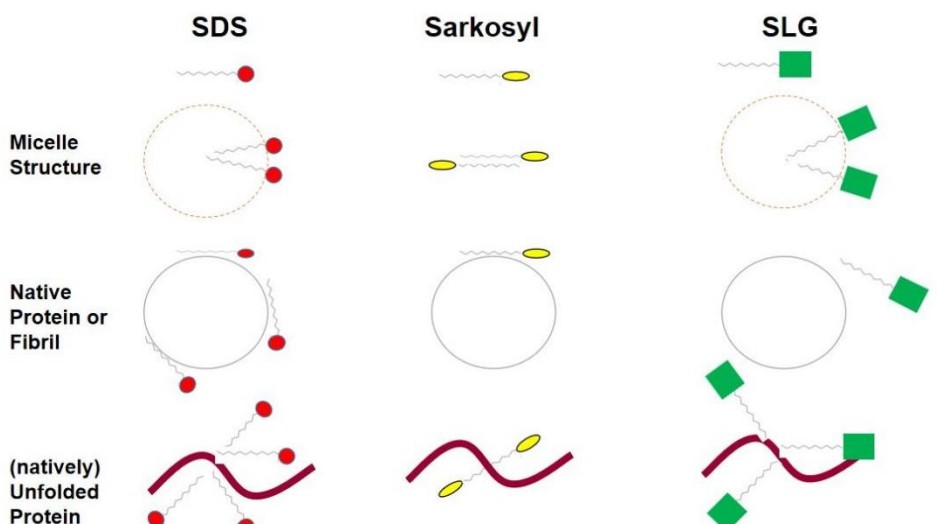

**Figure 2.** Schematic illustration of the micellar structure and binding mode of SDS, Sarkosyl, and SLG. Red circles (SDS), yellow ellipsoid (Sarkosyl), and green squares (SLG). Top row, the structure of the micelle. Middle row, the binding mode to native proteins (or fibrils). Bottom row, the binding mode to (intrinsically) native proteins. In these illustrations, the size of each structure does not reflect differences in the sizes of micelles and proteins.

The cleaning properties of SLG have been studied because of its cosmetic and cleansing applications [16]. When compared to Sarkosyl, SLG was slightly weaker in suppressing the surface tension of water, although both detergents are strong surface tension suppressors [17]. SLG is much stronger at stabilizing the foams than Sarkosyl despite their similarity in decreasing the surface tension of water. Interestingly, SLG is extremely weak in cleaning the greasy stains from surfaces below CMC, suggesting its weakness in hydrophobic binding as a monomeric detergent. Nevertheless, SLG can interact with hydrophobic aliphatic chains, as it enhances the solubility of alkyl-gallates, particularly for those with longer alkyl chains [18].

## 3. Effects on Native Proteins

The effects of Sarkosyl, SDS, and SLG were examined on the structure of native, folded interleukin-6 (IL-6) [19]. The protein was exposed to 0–2% detergent solutions and loaded onto a Superdex-75 gel filtration column run without the detergent. If the detergents dissociate from (or do not bind to) the protein without altering the protein structure, it shows native elution on the column. The peak area corresponding to the native protein was determined by the absorbance measurement. The recovery of native IL-6 from the 2% detergent solution is shown in Table 1. When diluted from 2% SLG upon gel filtration chromatography, the recovery was 100%. On the contrary, sodium lauroyl ether sulfate, used instead of SDS, resulted in 0% recovery. It is noted that sodium lauroyl ether sulfate was used here as an alternative to SDS, as the use of SDS in this experiment would give stronger denaturing effects on the protein and damage the column due to the strong (irreversible) binding of the SDS-denatured protein to the column. The recovery from 2% Sarkosyl was 38%, intermediate between lauroyl ether sulfate and SLG, indicating that the binding ability may be between lauroyl ether sulfate (and likely SDS) and lauroyl glutamate (SLG). These results for Sarkosyl and, likely, for SDS suggest binding to the native proteins, as shown in Figure 2.

An extensive analysis of the effects of SLG on native proteins was also carried out [20]. Circular dichroism (CD) was used to monitor the structural changes in IL-6 in the absence and presence of SLG. The CD spectrum of BSA in 2% SLG showed a distinct near UV CD spectrum with characteristic peaks and shoulders present in the native BSA but clearly with different intensities, indicating that 2% SLG alters the tertiary structure of BSA without

completely unfolding the structure. This, in turn, suggests that BSA can assume a native-like tertiary structure fold even in the presence of 2% SLG. The CD spectra of BSA in the presence of 0.1% SLG, generated whether diluted from 2% SLG or added to the native BSA, were essentially identical to the spectrum of native BSA, clearly indicating that BSA has a native fold in the presence of 0.1% SLG or regains the native structure when diluted from 2% SLG, as depicted in Figure 2 (middle row). Different from the above results with BSA, IL-6 showed a greater loss of the tertiary structure in the presence of 2% SLG, as determined by near UV CD analysis, indicating that the effects of SLG on the folded structure depend on the proteins. When SLG was removed, IL-6 completely regained the tertiary structure, as analyzed by CD, and consistent with the results of BSA. The tertiary structure of IL-6 was monitored using CD and fluorescence spectroscopy as a function of the SLG concentration, indicating that the structure change occurs around 0.2% SLG, and the structure is native below 0.1% SLG and unfolded above 0.5% SLG. Native gel electrophoresis indicated that SLG fully dissociates from BSA when contained only in the loading sample at 0.1–2%. However, IL-6 showed a doublet band, one of which is native, when 0.5% SLG is present in the loading sample, indicating that SLG at this concentration may still be bound to the non-native band of the doublet. Above 0.5% SLG, the native band was not observed, indicating that the dissociation of the bound SLG was not fast enough to generate the native IL-6 during the electrophoresis run time. To our knowledge, there appear to be no studies on the interaction of these detergents side-by-side with other proteins. Both IL-6 and BSA are small globular proteins, suggesting that it is of great interest to study larger proteins such as antibodies and protein complexes.

## 4. Effects on Protein Refolding

SDS is well known to bind to unfolded proteins, which is depicted in Figure 2 (bottom row). Sarkosyl has been shown to be non-denatured [21]. It was used to solubilize actin-binding proteins expressed as inclusion bodies in *E. coli* [22]. Sarkosyl was found to be able to extract active proteins, including the green fluorescent protein (GFP), tumor necrosis factor (TNF), and lymphotoxin from *E. coli* inclusion bodies [23]. Sarkosyl has been used to solubilize and refold proteins expressed recombinantly in bacterial cells [24,25]. A rather low concentration, 0.3–0.4%, of Sarkosyl was used to solubilize the polymerase sigma factor, which yielded a recovery of the folded protein greater than 50% [25]. Refolding was performed by the 10-fold dilution of the Sarkosyl concentration followed by dialysis. After dialysis, 0.01–0.02% Sarkosyl was still found to be trapped with the protein, most likely bound by the protein and, hence, un-dialyzable. It is speculated that Sarkosyl is bound by the unfolded sigma factor during solubilization, refolding during dialysis, and staying bound, though to a limited extent, but perhaps strongly, even after extensive dialysis. Sarkosyl was used to solubilize the granulocyte colony-stimulating factor (G-CSF) [26]. Native G-SCF has two disulfide bonds. When expressed in *E. coli*, G-CSF forms inclusion bodies. They were solubilized equally well using 6 M of guanidine hydrochloride or 2% Sarkosyl. Reverse-phase chromatography analysis showed a similar profile between these solubilizing solutions. G-CSF solubilized by these reagents was eluted between the fully reduced, unfolded structure and the native structure. Successful oxidation and refolding were achieved in the presence of 2% Sarkosyl with $CuSO_4$. In this study, the final products were analyzed using reverse phase chromatography, in which free (unbound-to protein) Sarkosyl was retained by the reverse phase column and eluted before the native G-CSF. It is evident that the strong binding of Sarkosyl by the reverse phase resin can completely be removed from the refolded G-CSF. It is not clear from this study how well Sarkosyl can be removed from the protein if a simpler process, such as diafiltration, gel filtration, and ion exchange chromatography, is used. A similar result was obtained in refolding a chymeric receptor binding domain (RBD) of SARS-CoV-2 fused to a DNA-binding CRM197 protein (i.e., CRM197-RBD fusion protein) [27]. The bacterial expression resulted in inclusion bodies, from which the chymeric protein was solubilized by 1% Sarkosyl. The Sarkosyl was

removed using a detergent removal resin, Amberlite XAD-4, leading to the formation of the functional structures of both the DNA binding protein and RBD.

The detailed protocol of application of Sarkosyl for the solubilization and purification of recombinant proteins from bacterial *E. coli* cells has been described, clearly showing its usefulness in protein refolding [13,28]. An interesting observation was made with glycerol kinase [29]. When expressed in *E. coli*, the glycerol kinase formed inclusion bodies. A cationic detergent CTAB (cetyltrimethylammonium bromide) at 0.5% solubilized the kinase without activity. SDS at 1% did the same but also without activity. Sarkosyl at 1% resulted in solubilization and an active enzyme. However, it was active only in the presence of 1% Sarkosyl, as its removal resulted in inactivation, perhaps due to the aggregation of the folded enzyme after removing Sarkosyl. This suggests that the bound Sarkosyl helps the kinase become soluble in an aqueous solution. As an Fc scaffold model, an attempt was made to express and characterize an antibody CH2-CH3 construct [30]. When the construct was expressed in *E. coli*, it was expressed in inclusion bodies. Sarkosyl (1.5%) was successfully used to solubilize the construct, and refolding into a native dimer was performed by dialysis. However, the recovery was rather low, which was ascribed to the persistent binding of the Sarkosyl, which hampered either the refolding process, chromatographic purification, or both. It is, thus, likely that Sarkosyl binds to the unfolded proteins, as depicted in Figure 2 (bottom row), and to the native proteins, though to a limited extent, as also shown in Figure 2.

SLG was used to solubilize or refold several proteins, including somatotropin. Native somatotropins contain two disulfide bonds. The solubilization and refolding of soma-totropins are reported in [31]. As an example, bovine somatotropin was solubilized in 2% SLG at an alkaline pH and refolded in the presence of oxidizing reagents via 2-fold dilution. A simple diafiltration was successfully used to remove SLG. SLG was also used to refold various proteins [32]. When expressed in *E. coli* cells, IL-6 was reduced and unfolded in insoluble inclusion bodies. It was solubilized by 2% SLG in the presence of disulfide exchange reagents. As described in the previous section, the CD analysis of native IL-6 showed that its structure is altered by SLG above 0.1%, clearly indicating that it can bind to the protein even below its CMC (~10 mM, see Table 1). After solubilization and incubation with 2% SLG, the solution was diluted 20-fold to 0.1% with 10 mM of the phosphate buffer at pH 7.0 and room temperature. The native, refolded IL-6 was identified with a high yield via ion exchange chromatography, indicating that dilution and ion exchange purification are sufficient to remove the bound detergent. Below 0.1%, SLG dissociates from the protein and leads to the formation of a native structure. It is interesting that Sarkosyl gave a low yield under the conditions used above. The use of arginine above 0.4 M in the phosphate refolding buffer (used for 20-fold dilution) resulted in greater refolding yield, suggesting that arginine assists the refolding and dissociation of SLG. The same solubilization and refolding technology was successfully applied to transglutaminase and the single-chain variable domain or Fab domain of antibodies [19,32,33].

The effects of temperature on the refolding and dissociation of SLG were examined using two single-chain antibodies (scFv) that also formed inclusion bodies [33]. Inclusion bodies containing these scFv were solubilized in 2.5% SLG and diluted with a phosphate buffer containing a redox agent at different SLG concentrations. The refolding and dissocia-tion of SLG were assessed by gel filtration using the Superdex-75 column. Refolding was independent of whether SLG was in micellar or monomeric forms, as the refolding yield was high below and above the CMC. Incubation at an elevated temperature, e.g., 45 °C, resulted in higher recovery, suggesting that higher temperature accelerated refolding. SLG at 2% was also used to refold an scFv-corestreptavidin fusion protein. The dilution of 2% SLG by 100-fold to 0.02% resulted in the refolding of corestreptavidin into a monomeric structure. The complete removal of SLG by dialysis led to the formation of tetrameric scFv-corestrepavidin fusion, which, in turn, suggested that a small amount of SLG was bound at 0.02%, most likely to the corestreptavidin moiety, preventing the formation of

the tetramer structure. These results clearly indicate that SLG does bind to the unfolded proteins, as shown in Figure 2.

## 5. Amyloid Fractionation by Sarkosyl or SDS

Tau is a member of "microtubule binding proteins (MAPs)", which bind to the surface of assembled tubulin subunits, i.e., microtubules, with a constellation of positive charges distributed through the tau structure [34,35]. Tau refers to natively (intrinsically) unfolded basic proteins with isoelectric points above nine and, hence, are positively charged at a physiological pH. The tau binds to the microtubules electrostatically and stabilizes the assembled structures. The phosphorylation of tau lowers net-positive charges and weakens the electrostatic interactions with the microtubules, making them less water-soluble. Neurofibrillary tangles are one of the major culprits in Alzheimer's disease and are formed by intra-cellular aggregates of abnormally hyperphosphorylated tau, a disease called "tauopathies". Tau aggregates have been characterized based on their solubility (or insolubility) in 1% Sarkosyl. The homogenization of brains in a buffer containing high salt (0.8 M NaCl) and 10% sucrose leads to the dissociation of tau from microtubules and helps the elimination of myelin and associated lipids [36]. The centrifugation of the suspension leads to the supernatant containing various forms of tau, to which the addition of 1% Sarkosyl with a reducing agent leads to the fractionation of the tau fibrils as Sarkosyl-insoluble aggregates [37]. It appears that the binding of Sarkosyl retains the "native"-like structures and hence antigenic epitopes of tau, which are lost when SDS is used instead of Sarkosyl and lead to the formation of insoluble aggregates that can be obtained via high-speed centrifugation [36]. One issue that is not clear is how 1% Sarkosyl binds to tau and generates (or retains) insoluble tau [38]. The 1% Sarkosyl was used not only in identifying and characterizing tauopathy-related insoluble tau fibrils but also the pathogenic fibrils derived from synucleinopathy α-synuclein and proteinopathy TDP-43. Thus, the solubility or, more directly, the insolubility of neuropathological fibrils by 1% Sarkosyl, or, as described below, 1–2% SDS, appears to be the hallmark of a technique for protein aggregates in neuron diseases, including Alzheimer's, Parkinson's and multiple system atrophy disease.

Cryoelectron microscopy analysis showed no major effects of the Sarkosyl treatment on the structure of Aß and tau fibrils [39], although Sarkosyl-untreated Aß, i.e., the ultra-centrifugation supernatant of Tris-buffered saline extracts of Alzheimer's disease brain, showed shorter fibrils that clumped less, suggesting that Sarkosyl may enhance fibril association [40]. Unfortunately, no data are available on the effects of SLG on fibrils. Based on its mild detergent properties, it would be of great interest to examine its ability to bind to fibrillar aggregates and other proteins and separate the fibrils from SLG-sensitive proteins and complexes.

SDS has also been used to fractionate amyloid fibrils, taking advantage of the SDS-resistant properties of amyloid fibrils. Prion amyloids were separated by treating cell lysates with 2% SDS at 37 °C, which enabled proteins other than the amyloids to solubilize; however, care must be exercised not to overheat to avoid the disaggregation of the amyloid structures. Instead of 2% SDS, 1% Sarkosyl may be used [41]. The use of Sarkosyl was found to detect a wider range of amyloids than SDS due to Sarkosyl's weaker detergent capability and the susceptibility of amyloids to SDS, although Sarkosyl may retain non-amyloid complexes. Polyglutamine (polyQ) fibrillar aggregates can be readily fractionated by SDS treatment, as polyQ is SDS-resistant [42]. As noted above, there may be a disadvantage to using a weak Sarkosyl detergent, which can retain certain non-amyloidogenic complexes, resulting in contaminating proteins in amyloid fractions. A different combination of temperature and SDS concentration, e.g., 1% SDS and room temperature, can also be applied to retain and purify amyloid aggregates [43]. Utilizing the detergent-resistance properties, previously unidentified amyloid-like structures have been discovered in different organisms, e.g., such eukaryotes as yeast and plant seeds [44–47]. Amyloid-like proteins were obtained from pea seeds as 1% SDS-resistant proteins and identified as vicilin.

## 6. Cell Lysis

Many in vitro cell biology experiments require cell lysis, as depicted in Figure 3, which releases soluble and membrane proteins and may or may not preserve the native integrity of cellular components. One such procedure is called decellularization, which is often performed on tissues for xenografts. For example, corneal transplantation is the only option to cure corneal opacities using decellularized corneas and is performed with different decellularized protocols [48]. Decellularization removes xenogenic cells and antigens, which can cause rejection, for tissue transplantation. SDS, in combination with nucleases, has been used in traditional decellularization, as this detergent can remove nuclear and cytoplasmic components from various tissues, including the dermis, kidney, and lung [49]. However, SDS can damage biologically essential components that constitute the extracellular matrix (ECM). Thus, SDS was compared with Sarkosyl, Triton X-100, or SLG in combination with nucleases for the decellularization of porcine corneas that may be applied to human patients [50]. The results are clear that SLG was better than the other three detergents in preserving the transparency of the cornea and removing residual cellular nuclei and other components from the porcine cornea. In addition, SLG resulted in the regular alignment of corneal stromal fibrils, while another three detergents resulted in the disordered alignment and retention of residual nuclei. These results indicate that SLG is superior in removing xenogenic cells and antigens and preserving the transparency and ultrastructure of corneal stroma.

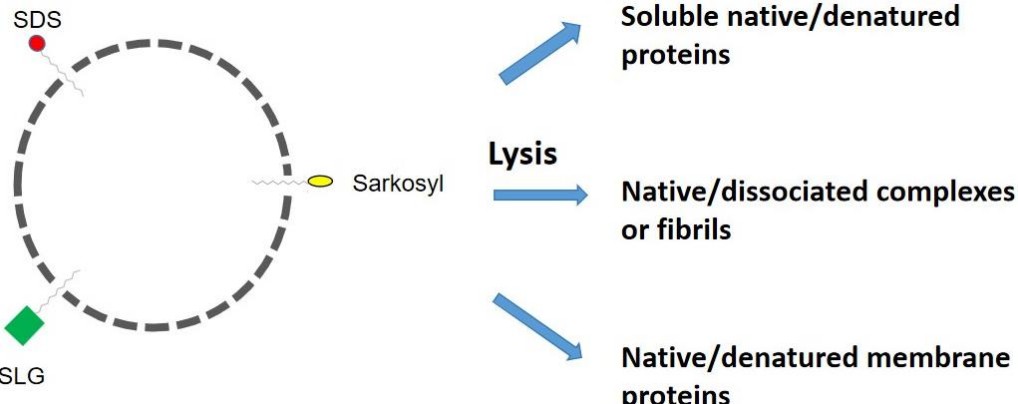

**Figure 3.** Schematic illustration of a detergent binding to membranes and its effects on proteins released from lysed cells. SDS, Sarkosyl, and SLG are shown to penetrate cell membranes through their aliphatic chains. Their head groups are in the bulk solvent, as indicated by the red circle (SDS), yellow ellipsoid (Sarkosyl), and green square (SLG). These detergents are expected to affect soluble, complex, or fibrillar membrane proteins differently.

SDS has been extensively used to lyse cells under denaturing conditions for proteomic analysis, namely, cell lysis above the CMC, followed by the removal of SDS via protein precipitation, washing steps, and protease digestion before mass analysis [51]. While this is highly effective in cell lysis and proteomic analysis, the extracted proteins from the cells are inherently denatured and, hence, may not be suitable for functional proteomics. However, there is an advantage of this denaturing/dissociating ability of SDS for the isolation of SDS-resistant fibrils, as described below and depicted in Figure 3. An even harsher form of cell lysis is to use SDS in combination with heating, which guarantees complete cell lysis and protein denaturation for immunoprecipitation using antibodies that react only with linear peptide sequences [52].

Sarkosyl at 1% has been used to aid with cell lysis, thereby removing nucleic acids from purified proteins [53]. They also showed that a higher Sarkosyl concentration, even 10%, is effective at solubilizing multiple proteins from inclusion bodies, conferring folding into the native structure and speculating the mechanism of solubilization via the encapsulation of native-like proteins trapped in inclusion bodies, assuming that these inclusion bodies

are made of native or native-like structures that are made insoluble due to a high protein concentration or the presence of nucleic acids, lipids or other contaminants. Sarkosyl at 2.5% was used to increase the permeability of *E. coli* cell membranes [54].

The subcellular fractionation of *Shewanella oneidensis* proteins was successfully obtained with 0.5% Sarkosyl [55]. After mechanical cell lysis, the subcellular complex of the lipid–protein complex was solubilized by Sarkosyl for proteomic analysis via mass spectrometry. Identification of bacterial Rhodococcus strains was performed using a lysis protocol made of Sarkosyl and lysozyme, combined with freeze–thaw, followed by PCR analysis [56]. The lysis of *E. coli* cells and solubilization of proteins by Sarkosyl was carried out for the expression of the ecdysteroid receptor (EcR) and ultraspiracle (USP) together. The purification of these proteins resulted in the expression of the functional heterodimer and homodimers of both EcR and USP [57]. The combination of Sarkosyl-based cell lysis and a different *E. coli* strain resulted in the expression of soluble actin, which otherwise formed inclusion bodies [58].

Although these detergents may be equally effective in cell lysis, they are different in their effects on proteins released from lysed cells. How these detergents affect the soluble, complex, and membrane proteins, as shown in Figure 3, depends on the detergent concentration and such solution conditions as pH and temperature. For protein purification, SDS is most unfavorable at obtaining native soluble proteins and native molecular complexes from the cytoplasm, nucleus, and membrane proteins. The use of Sarkosyl or SLG may require adequate procedures for the purification of native proteins, as they add negative charges when bound to the proteins and may dissociate native complexes, although their dissociating capacity is weaker than SDS. The extracellular domain structure of the membrane proteins is most likely destroyed by SDS. To isolate and purify membrane proteins, however, it is essential to use detergents to stabilize their membrane-spanning domains. The detergents most frequently used are non-ionic (e.g., Triton X-100) or zwitterionic (e.g., CHAPS) detergents or mild ionic detergents (e.g., sodium cholate), which are milder than Sarkosyl, SLG, and SDS [59,60]. Depending on the subsequent experiments, the detergent used to lyse the cells and extract membrane proteins may need to be removed or replaced with another detergent, meaning that the ability of the detergent to dissociate from the solubilized proteins plays a key role in the purification and characterization of proteins.

## 7. Mechanism

Here, we do our best to interpret the effects of these three anionic detergents. One of the most critical questions is the effects of SDS, Sarkosyl, and sodium lauroyl glutamate (SLG) on membranes and macromolecules during cell lysis, decellularization, and purification. These three detergents may be expected to penetrate the cell membrane equally well because of their identical aliphatic chains (see Figure 3). However, they are different in their effects on proteins released from lysed cells. Among the three detergents, SDS is strongest in its ability to bind to and denature proteins. Why is SDS stronger than the other two detergents? They have an identical aliphatic chain and differ mainly at the charged group, as is clear from the different molecular weights (see Table 1) depicted in Figure 2 (different sizes of the head groups). One of the mechanisms that is caused by the different charged groups may be the uniqueness of the sulfate group in SDS. Sulfate salts, e.g., sodium, ammonium, and even guanidinium sulfate, are known as strong salting-out salts [61–63]. Although pure speculation, the sulfate group in SDS may salt out itself (SDS) either intra-molecularly or inter-molecularly at a local high SDS concentration when bound by proteins. Namely, hydrophobic interactions between the aliphatic chain of SDS and the non-polar surface of the protein molecules are enhanced by their own sulfate group. By the same token, it is expected that these head groups are different in size and hydration and play a role in determining their differences. SLG has the largest head group, which may destabilize its binding as a cluster due to steric hindrance. The sulfate group in SDS is perhaps the most hydrated and, hence, the most effective in stabilizing SDS-bound structures. Namely, SDS not only masks the hydrophobic surface of the proteins but also makes the bound structure

more hydrated, stabilizing the denatured and dissociated structures. There may be other factors involved in the different strengths of these detergents, which require more studies.

The next question is their effects on denatured proteins. These three detergents are used to solubilize proteins from inclusion bodies. One of the mechanisms for their solubilization of proteins from inclusion bodies is the dispersion or dissociation of contaminants that are trapped in inclusion bodies and possibly bound by the proteins. Lipids and nucleic acids may be effectively dispersed and dissociated from the proteins by the detergents. For example, Sarkosyl helps dissociate and remove lipopolysaccharides (LPS) from Gram-negative bacteria such as *E. coli* and *Chlamydia* [64]. At elevated temperatures, LPS undergoes thermal transition, making its binding weaker, leading to its replacement by Sarkosyl. The binding of Sarkosyl then prevents the protein from aggregation [65]. The dispersion of lipids by the detergents can be explained by the binding of the aliphatic groups of the detergents. Above the CMC of the detergents, lipids may form mixed micelles with the detergents, although Sarkosyl does not appear to form micelles by itself, even above the CMC. However, Sarkosyl may be able to bind to the lipids through its aliphatic chains and increase the solubility of the lipids. Nucleic acids may be bound by the detergents through aromatic/hydrophobic interactions with the nucleobases [66]. However, it is unlikely that dispersing contaminants from inclusion bodies is sufficient to make the denatured proteins soluble. These denatured proteins in the inclusion bodies aggregate to a different extent depending on the proteins expressed in inclusion bodies.

SDS has been shown to bind to denatured proteins as stable SDS–protein complexes. As shown in Figure 2, SDS binds to an (unfolded) polypeptide as micellar forms, masking its non-polar surface and, more importantly, adding negative charges and, possibly, hydration to the polypeptide. SLG, which also forms micelles (see Table 1), most likely binds to proteins in a similar manner but less stably than SDS micelles due to the bulky head group of SLG. Sarkosyl may also bind to proteins similarly, but perhaps not in a micellar manner. It is also possible that Sarkosyl binds to proteins molecularly, also masking non-polar surfaces and adding negative charges. Refolding from the detergent-solubilized proteins can be effectively undertaken both for SLG and Sarkosyl, which may be ascribed to unstable detergent binding to the proteins.

An attempt was made to use Sarkosyl in place of SDS in gel electrophoresis [67,68]. Two purified recombinant proteins, one with an isoelectric point of 4.64 and molecular weight of ~17,000 and another with an electric point of 8.0 and a molecular weight of ~30,000, were subjected to Sarkosyl-PAGE, containing 0.05% Sarkosyl, which is below the CMC. Unfortunately, the running pH was not reported in that paper [54], although the pH of the loading sample was 6.8. These proteins showed the electrophoretic mobility of the monomer expected from their molecular weight, most likely due to Sarkosyl binding proportionally increasing with the molecular size of the proteins. These proteins showed a doublet band due to dimerization, as expected from their self-association properties. NMR experiments indicated the retention of the native fold for several proteins and the perturbation of surface residues. These results clearly indicate the binding of Sarkosyl on the native proteins during electrophoresis and the NMR experiment. A successful Sarkosyl solubilization (even 10%) followed by dilution refolding has also been shown for the maltose-binding protein expressed as a fusion protein to glutathione S-transferase (GST) in inclusion bodies [69]. It was shown, however, that reducing the Sarkosyl concentration alone does not restore the native structure of the GST; instead, a combination of dilution with the addition of Triton X-100 and CHAPS resulted in the renaturation of the GST portion. This was ascribed to the binding of Sarkosyl to the GST, which disabled binding to the glutathione column. Those bound to Sarkosyl can then be removed from the glutathione binding site via the addition of milder detergents (Triton X-100 and CHAPS). Thus, Sarkosyl appears to persistently bind to proteins even after refolding, as described earlier.

The most frequently employed solubilization methods are the use of urea and guanidine hydrochloride, which not only fully denature the proteins but also lead to effective solubilization by cleaving inter-molecular and intra-molecular hydrogen bonding and

hydrophobic interactions. Their binding to the proteins is generally weak compared to the detergents. The binding of these detergents occurs in the mM concentration range, while urea and guanidine hydrochloride require an M concentration range. After refolding, they can be readily dialyzed out. Which of the detergents or the denaturants is more effective in refolding proteins depends on the proteins. However, it seems clear that the detergent process is more cost-effective.

Detergents are frequently used for chemical cell lysis procedures. Since SDS, Sarkosyl, and SLG all have the same aliphatic chain, they may be equally effective in penetrating cell membranes, as shown in Figure 3, and disrupting the membrane structures. SDS has been used as a means of denaturing cell lysis, as this detergent most likely denatures the proteins extracted from the cells. Namely, excessive SDS binds to the proteins and denatures them. However, it may be effective at solubilizing membrane proteins but can still denature their extracellular domains. Sarkosyl may similarly disrupt the membrane structures and release cellular proteins. Although it may bind to the proteins, the extracted proteins are most likely functional. There is no report on the use of SLG for cell lysis, but one of us (Tsutomu Arakawa) tested SLG and observed functional proteins when extracted from the cells using SLG. It would be of great interest to try this detergent for cell lysis because of its weak protein-binding properties and non-denaturing nature.

The last question is how Sarkosyl or SDS separates these insoluble neuropathogenic fibrils. Taking tau as an example, this protein is natively unfolded and bound to microtubules, as described earlier. It undergoes post-translational modifications, including phosphorylation, which, in turn, leads to conformational changes and a series of oligomerization, eventually leading to the formation of filaments and fibrils [70]. The native proteins and their oligomers are actually more likely to be less folded than the late-stage filamentous and fibrillar structures and may present binding sites for Sarkosyl or SDS. The early-stage and late-stage oligomers were both characterized as granules using atomic force microscopy, dynamic light scattering, and CD spectroscopy [71]. The early-stage oligomer granules are Sarkosyl-soluble, smaller in size, and have extended structures, while late-stage oligomer granules (closer in structure to fibrils) are Sarkosyl (or SDS)-insoluble and β-sheet structures. Thus, the late-stage samples may contain various forms of tau or other neuropathogenic proteins. The binding of Sarkosyl to these less-folded species may make them more soluble and facilitate their separation from less-soluble (Sarkosyl-insoluble) forms. In this sense, it also presents binding sites for SDS. However, SDS can cause structural changes in fibrils that are different from Sarkosyl, which may preserve the fibril structures more effectively. SLG may also be effective in keeping the fibrillar structure intact, although its binding to the natively unfolded structure is not known. It should be noted, nevertheless, that SLG can bind to the denatured proteins expressed in inclusion bodies. The binding of Sarkosyl to natively unfolded proteins may make them more soluble and, thereby, facilitate their separation from Sarkosyl-insoluble fibrils, which may bind Sarkosyl to a limited extent and hence may not be solubilized and converted to the natively unfolded state.

## 8. Conclusions

Here, we review the properties and effects of three anionic detergents, SDS, Sarkosyl, and Sodium lauroyl glutamate (SLG), which have been used in molecular biology research. SDS is effective in cell lysis, the solubilization of proteins, and the purification of SDS-resistant fibrils, but inherently denatures proteins and disrupts native complexes. Sarkosyl binds to unfolded proteins and, most likely, to native proteins, though to a limited extent, and can be used for protein solubilization and the separation of soluble proteins from Sarkosyl-insoluble neuropathological protein fibrils. SLG is mostly mild as a detergent and can be used to refold proteins, although its use as a molecular biology research reagent has been limited. We hope that this review promotes research on its use in cell lysis, the protein refolding of inclusion bodies, and the characterization of protein fibrils.

**Funding:** This review article received no external funding.

**Data Availability Statement:** No new data were created or analyzed in this study. Data sharing is not applicable to this article.

**Acknowledgments:** We thank Daisuke Ejima for his insightful suggestions and kind help.

**Conflicts of Interest:** Tsutomu Arakawa and Y.K. were formerly affiliated with the for-profit company Alliance Protein Laboratories but currently have no conflicts of interest. Teruo Akuta is an employee of the for-profit company Kyokuto Pharmaceuticals. T.N. is a staff member at Sophia University and has no conflicts of interest.

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
