# Peer review of "Sodium Dodecyl Sulfate Analogs as a Potential Molecular Biology Reagent"

_cimb, doi:10.3390/cimb46010040_

Round 1

Reviewer 1 Report

Comments and Suggestions for Authors

The manuscript is devoted to a comparative analysis of the properties of three anioinic detergents, SDS, Sarkosyl and sodium lauroylglutamate (SLG). The authors describe in detail the effects of these detergents on native proteins, protein non-refolding, and cell lysis. In general, the presented data can be useful for a wide range of researchers. At the same time, there are certain additions and comments to the text of the article.

General remarks:

The authors write that it is Sarkosyl used for characterization of neuropathological protein fibrils and solubilization of proteins. Meanwhile, in recent years, much more work has been devoted to the use of SDS to isolate amyloid fibrils and to identify the proteins that form these fibrillar structures. SDS does dissolve most protein complexes, fibrils and aggregates, however all known amyloid fibrils are resistant to treatment with 1-2% SDS at room temperature (Cold Spring Harb Protoc. doi: 10.1101/pdb.prot089045; PLoS One. doi: 10.1371/journal.pone.0116003; Methods Protoc. doi: 10.3390/mps6010016). Based on this property of SDS, universal methods for proteomic screening and identification of amyloids have been developed. Fibrils that are resistant to SDS treatment are separated from other proteins, and at the next stage, the proteins included in the fibrils are identified using mass spectrometry. These proteomic methods have enabled the identification of previously uncharacterized amyloids in yeast, plants, and mammalian brain neurons (Curr Genet. doi: 10.1007/s00294-017-0759-7; PLoS Biol. doi: 10.1371/journal.pbio.3000564; Sci Rep. doi: 10.1038/s41598-019-55528-6). Amyloid fibrils are also resistant to Sarkosyl, but other protein complexes are also resistant to treatment with this detergent. For example, the 20S catalytic core of the yeast proteasome is insoluble in 3% Sarkosyl at room temperature (PLoS One. doi: 10.1371/journal.pone.0116003). In this regard, Sarkosyl is worse to use for searching and identifying amyloid fibrils. I recommend taking this note into account in all sections of the article, starting with the Abstract. The title of the subsection “Tau fractionation by Sarkosy” should also be changed to discuss the use of SLS and Sarkosyl for the study of amyloid fibrils.

Minor remarks:

1.       Abstract.  “Sarkosyl is used for characterization of neuropathological protein fibrils and solubilization of proteins”.

SDS is also used to study fibrillar proteins to an even greater extent.

 2.       Page 6. “Thus, the solubility or more directly insolubility of neuropathogi-cal fibrils by 1 % Sarkosyl appears to be a hallmark of technique to be used for protein aggregates in neuron diseases, including Alzheimer’s, Perkinson’s and multiple system atrophy disease.”

 A similar method has already been developed, but using SDS (see above). In addition, typo in ‘Parkinson’s’.

 3.       Page 6. “Unfortunately, no data are available on the effects of SLG on fibrils. Based on its mild detergent properties, it would be of great interest to examine its ability to bind and separate fibrillar aggregates”.

This detergent is unlikely to be suitable for separating fibrillar aggregates. Even SDS is not able to separate some amyloid aggregates into fibrils (Biophys Res Commun. doi: 10.1006/bbrc.2000.3682)

 4.        Page 7. “Combination of Satkosyl-lysis and a different E. coli strain resulted in expression of soluble actin, which otherwise formed inclusion bodies”.

 Strange wording. The sentence should be rewritten. Note, typo in “Sarkosyl”.

 5.        Page 9   Figure 3 provides little information. I recommend removing this Figure.

Reviewer 2 Report

Comments and Suggestions for Authors

 This study examines the physical properties of two SDS analogs: sodium lauroyl glutamate (SLG) and sodium lauroyl sarcosine (Sarkosyl). The role of the detergents is illustrated convincingly for several proteins. Still, the limited number of studied proteins significantly reduces the enthusiasm of the reviewer and probably the interest of a wide audience. While the manuscript cites the most relevant work in the field, the information is often provided as a list of events rather than an integrated perspective.

 Specific comments.

-SLG and Sarkosyl have rather similar structures but their properties in terms of dissolution of aggregates or membrane destabilisation are different. Unfortunately, very little information is available regarding the mechanisms involved.

 - It is somehow surprising to read that Sarkosyl aggregates as a dimer. How would an acyl chain accommodate such an aqueous environment.

- Effect on native proteins-Why to limit the discussion to Il-6, BSA. Extension to a larger number of soluble and membrane proteins would have been quite informative.

 - The study focuses heavily on the dissolution of bacterial aggregates but ignores the potential of SFG and Sarkosyl in purifying and refolding membrane proteins.

Minor point

- Figure 3 does not contribute to a better understanding of the detergent-membrane interaction and can be deleted.

Round 2

Reviewer 1 Report

Comments and Suggestions for Authors

Minor remarks:

1.       The use of Sarkosyl was found to detect a wider range of Aß amyloids than SDS due to Sarkosyl’s weaker detergent capability and a susceptibility of Aß to SDS, although Sarkosyl may retain other non-amyloid complexes. Polyglutamine (polyQ) fibrillar aggregates can be readily fractionated by SDS-treatment, as polyQ was SDS-resistant [42].

The definition of "Aß amyloids" is meaningless. Amyloid peptide beta (Aβ) is only one of many peptides and proteins that are capable of forming amyloid fibrils (e.g. the fibrils with cross-β-sheet structure). Insert "amyloids" for "Aß amyloids".

2.       Utilizing the detergent-resistance properties, previously-unidentified amyloid-like structures have been discovered in different organisms, including yeast [44], eukaryotes [45, 46] and plant seeds [47].

Yeast and plants are eukaryotes. The sentence needs to be corrected.

3.       Amyloid-like proteins were obtained from pea seeds as 1 % SDS-resistant proteins and identified as vicilin, ferritin and dehydrin. The recombinant vicilin showed resistant to 2 % SDS and formed amyloids positive with amyloid-specific fluorescent Thioflavin T dye when incubated in PBS.

The main results of the paper cited by the authors are presented incorrectly. Amyloid properties in plant seeds have been shown only for vicilin. Ferritin and dehydrin and more than 30 other proteins form SDS-resistant aggregates in pea seeds, but this does not mean that they are amyloid-like proteins. The amyloid properties of vicilin in vivo and in vitro have been demonstrated using a number of approaches. Binding to Thioflavin T is only one of the evidence presented. I doubt it's worth writing about these details.

4.       Depending on the subsequent experiments, the detergent used to lyse the cells and extract membrane proteins may need to be removed or replaced with another detergent, meaning that the ability of the detergent to dissociate from the solublized proteins plays a key role in the purification and characterization of the proteins..

Remove an extra dot at the end of the sentence.

Author Response

  1. A-beta was removed. A-beta amyloid changed to amyloid.
  2. The sentence was revised to show yeast and plant as eukaryote.
  3. Only vicilin was described and the sentence was shortened.
  4. One comma removed.

Reviewer 2 Report

Comments and Suggestions for Authors

The authors have answered all my queries by modifying the manuscript or in their written responses.

Author Response

Thank you for accepting the revisions.